# More intense heatwaves under drier conditions: a compound event analysis in the Adige River basin (Eastern Italian Alps)

Marc Lemus-Canovas<sup>1</sup>, Alice Crespi<sup>1</sup>, Elena Maines<sup>1</sup>, Stefano Terzi<sup>1</sup> & Massimiliano Pittore<sup>1</sup> Center for Climate Change and Transformation, Eurac Research, Bolzano-Bozen, 39100, Italy

5 Correspondence to: Marc Lemus-Canovas (marc.lemusicanovas@eurac.edu)

**Abstract.** The Adige River basin has been affected several times in recent years by concurrent very hot and dry conditions. In summers 2015, 2017, and more recently in 2021-2022, severe hydrological droughts compounded and cascaded with wildfire and heatwave events. The chained effect of snow deficit in winter, higher-than-normal temperatures in early spring and heatwaves during summer caused multiple drought impacts. Despite the severe consequences, the role of observed climate change in exacerbating the intensity of the drivers leading to specific hot and dry events and their potential impacts in this region remains poorly understood.

A ranking of compound drought and heatwave events (CDHW) occurring in the Adige River basin between 1950 and 2023 was built using E-OBS precipitation and temperature observations. The ranking was based on a composite index considering both event intensity and the spatial extent of the affected area, derived from the Standardised Precipitation Index at a 6-month scale (SPI-6) and a heatwave definition based on daily maximum temperature (TX). The major 2022 CDHW event, which caused severe environmental and societal impacts in the Adige River catchment, stood out. Occurring in late spring (10–28 May 2022), it ranked fifth out of 119 events detected since 1950 and was the most intense CDHW event in the past 15 years. As one of the most recent and severe CDHW events, the 2022 event was selected for an in-depth characterisation and a climate change attribution analysis of both its meteorological drivers and hydrological impacts.

The changing characteristics of CDHW events similar to that of May 2022 were investigated through a flow-analogue attribution approach based on the reconstruction of its atmospheric conditions using ERA5 geopotential height at 500 hPa. By comparing May 2022 CDHW flow analogues from 1951–1980 (low anthropogenic forcing) and 1992–2021 (moderate-high anthropogenic forcing), we found that heatwaves comparable to the one in May 2022 are now significantly hotter –by 1-4°C– than historical analogues and occur in a much drier context, characterised by pronounced precipitation deficits.

These conditions, along with earlier snowmelt and persistent precipitation deficits, might have exacerbated river flow

reductions and water stress in recent years. Also, shifts in the timing of a CDHW event were found to significantly influence the severity of its potential consequences.

However, extracting a reliable signal of future changes in the characteristics of CDHW events from climate projections remains challenging. Based on flow-conditioned analogues of the May 2022 event from 25 EURO-CORDEX simulations, more than half of the models failed to reproduce the observed sign of change in temperature and drought conditions. Unconditioned reconstructions showed closer agreement with observations, particularly for temperature patterns, but critical aspects such as the magnitude of the changes remained underestimated.

# 1 Introduction

40

60

The Alpine region is often referred to as the water tower of Europe due to its crucial role in sustaining water supply for downstream areas (Viviroli et al., 2007). Given its high sensitivity to climate change, any modification in climatic conditions can trigger significant effects on the availability and reliability of water resources.

Compound hot and dry extremes are internationally recognized for their impactful and wide effects on vegetation (Baronetti et al., 2023), human health (Morlot et al., 2023) and agriculture (Zanotelli et al., 2022) leading to larger impacts than the sum of the individual extreme effects (Hao et al., 2022; Zscheischler et al., 2020). Hot and dry conditions can trigger cascading processes such as wildfires (Feng & Sun, 2024; Fan et al., 2023; Grillakis et al., 2022), hydrological droughts (Hao et al., 2022; Feng et al., 2022), and crop failures (Brás et al., 2021; Ribeiro et al., 2020), leading to significant socio-economic losses. Compound hot and dry extremes can be influenced by both local thermodynamic processes and large-scale atmospheric circulation patterns, which interact at different spatial and temporal scales. Prolonged heatwaves in the central Mediterranean region are typically driven by the persistent presence of a high-pressure subtropical ridge over southern Europe (Sousa et al., 2021), which pushes warm air masses from northern Africa towards the northern Mediterranean basin. In addition, the associated subsidence enhances adiabatic warming and increases surface radiative fluxes, favouring the maintenance of extreme near-surface temperatures (Miralles et al., 2014; Zschenderlein et al., 2019). This atmospheric circulation pattern results in prolonged periods of high temperatures and reduced precipitation (Lemus-Canovas et al., 2025), thereby exacerbating pre-existing drought conditions (Bakke et al., 2023; Faranda et al., 2023). Moreover, the co-occurrence of warm air masses over already dry soils triggers positive feedback mechanisms, where the lack of soil moisture reduces evaporative cooling, further intensifying extreme temperatures (Miralles et al., 2014; Lemus-Canovas et al., 2024a).

Recent studies showed that compound hot and dry events have been increasing throughout the last decades in Europe and this trend is projected to continue with global warming (Felsche et al., 2024). Some of severe extreme hot and dry episodes which recently affected large parts of Europe, such as in 2003 (Fink et al., 2004), 2018 (Stephan et al., 2021) and 2022 (Tripathy and Mishra, 2023), showed also negative impacts in Alpine regions, especially in their lowlands. Given the accelerated warming of the Alps and the heterogeneity of their ecosystems and climatic regimes, anomalously dry and warm conditions are expected to increasingly affect Alpine areas in next future (Kotlarski et al., 2023). Understanding the main climatic drivers triggering the occurrence of compound hot and dry events is crucial to improve event predictions and better assess climate change effects. In particular, attribution analyses of specific events to anthropogenic climate change and detailed evaluation of potential primary causes of their intensification can help future impact modelling and serve as an essential resource for adaptation planning.

However, studies identifying compound drought and heatwave (CDHW) events and attributing their observed changes in the Alpine context, especially at catchment or sub-catchment scale, remain scarce. Existing works have primarily focused on large-scale phenomena affecting continental regions (Faranda et al., 2022; Stott et al., 2016), often overlooking the complex interactions between climate drivers and hydrological responses in mountainous environments. In this context, flow-

analogue approaches have recently emerged as a powerful tool to attribute extreme events, as they enable the separation of the thermodynamic signal of climate change from circulation-related variability (Yiou et al., 2014; Vautard et al., 2016; Franda et al., 2024). These methods have been successfully applied to a range of extremes, including heatwaves (Jézéquel et al., 2018; Lemus-Canovas et al., 2024b; Trigo et al., 2025) and heavy precipitation (Thompson et al., 2024; Ginesta et al., 2023), highlighting their ability to link large-scale atmospheric dynamics with local impacts. Such an approach is particularly promising for Alpine catchments, where complex orography and hydrological processes complicate the detection and attribution of extreme events. These challenges are closely tied to the unique physical processes that characterise mountain environments, which further complicate climate change attribution.

Indeed, mountain catchments present unique challenges for attribution studies due to the strong influence of snow dynamics (Brunner et al., 2023; Jenicek et al., 2016; Van Loon et al., 2015), elevation-dependent warming (Pepin et al., 2015), and interconnected mechanisms between atmospheric water demand and water availability (Mastrotheodoros et al., 2020), factors that are often underrepresented in large-scale studies. In addition, few studies have evaluated the performance of regional climate models (e.g., EURO-CORDEX) in applying attribution techniques, even in large-scale settings (Vautard et al., 2022). As a result, the accuracy of these models in reproducing circulation-driven changes in CDHW conditions in Alpine environments remains unexplored. This lack of evaluation underpins the need to assess the reliability of regional climate simulations for projecting future CDHW events, a crucial step in ensuring robust adaptation strategies for mountain catchments. However, even though EURO-CORDEX simulations provide added value in mountainous areas compared to CMIP5/6 simulations thanks to their higher resolution (Bilbao-Barrenetxea et al., 2024), they also present inherent limitations, such as their horizontal resolution of  $0.11^{\circ}-0.44^{\circ}$ , which can be insufficient to capture local-scale processes in complex Alpine terrain (Kotlarski et al., 2023; Giorgi et al., 2016), as well as the uncertainties introduced by convection and land-atmosphere parameterizations that strongly affect extremes (Ban et al., 2014). The Adige River basin in north-eastern Italy is one of the major Italian catchments and was one of the Alpine basins mostly affected by some recent high-impact hot and dry conditions, e.g., during the summers of 2015 (European Drought Observatory, 2015), 2017, and 2022 (Colombo et al., 2023). Due to the heterogeneity of its ecosystems, the complex socio-economic context, and its crucial interconnections with downstream areas, the Adige River basin proved to be particularly sensitive to the effects of CDHW events, when severe hydrological droughts occurred also in combination with wildfire events.

In particular, the winter of 2021/2022 was characterized by a lack of snowfall, followed by a rainfall deficit in early spring and above-average temperatures (Koheler et al., 2022). These conditions led to decreased soil moisture and reduced river streamflow and water availability in artificial basins and lakes (Avanzi et al., 2024). From early summer 2022, severe water-use restrictions affected multiple sectors in the Adige basin, including domestic supply, agriculture, and hydropower. Some hydropower plants were temporarily shut down and the minimum ecological flow in parts of the Adige River basin was reduced (Il Dolomiti, 2022). The insufficient availability of water for irrigation led to a shrinkage in apple production and quality in the upper basin (Alto Adige, 2022). Furthermore, a series of three consecutive heatwaves hit the area between May and September, leading to several weather alerts issued by the Civil Protection Agency and an estimated 309 attributed

fatalities among the elderly (≥ 65 years) population in the towns of Bolzano (40 attributed fatalities), Trento (28), Verona (23), Vicenza (22), Padova (96) and Venice (100) (Italian Ministry of Health, 2025).

In this study, we aimed to (i) identify the most severe CDHW events in the Adige River catchment over recent decades using a composite indicator that accounts for both their spatio-temporal extent and magnitude; (ii) assess the relative intensity of the hot and dry event of 2022 with respect to the other identified CDHW episodes and attribute observed changes in the meteorological drivers exacerbating the severity of this recent event, specifically by focusing on how regional warming trends influence atmospheric circulation patterns and near-surface related variables; (iii) evaluate the impacts of changes in snow dynamics associated with CDHW conditions on streamflow patterns; and (iv) evaluate the ability of EURO-CORDEX regional climate models to reproduce observed changes in drought and heat conditions, in terms of both sign and magnitude, in order to assess their suitability for extending attribution analyses to future climate projections in mountainous regions.

#### 110 **2. Study area**

105

The Adige River is the second longest river in Italy (410 km) and the third one for its overall basin area (12200 km<sup>2</sup>, Navarro-Ortega et al., 2015). It starts in the mountainous province of Bolzano in the north (62% of the overall basin), crosses part of the province of Trento in the south (29%), the provinces of Verona, Vicenza, Padova, Rovigo and Venezia in the Veneto region (all of them summing to 9%) and ends in the Adriatic Sea (Figure 1). In the last part of the river, the basin consists of the riverbed itself, laying at a higher elevation than the adjacent floodplains and receiving no contributions from any tributaries. Almost half of the overall residential population of the basin is concentrated in the main urban areas in the valley bottoms such as Bolzano-Bozen, Trento, Rovereto and Verona with values in 2001 pointing to 1,316,909 inhabitants (Distretto Alpi Orientali, last access: 10/03/2025).

Figure 1. Map of the Adige River basin with elevation data, administrative borders and locations of the selected snow depth and streamflow gauging stations used in the study. Elevation data: Shuttle Radar Topography Mission (SRTM) Global, distributed by OpenTopography, <a href="https://doi.org/10.5069/G9445JDF">https://doi.org/10.5069/G9445JDF</a> (accessed: 2025-02-25).

Climate is affected by the basin orography and is characterised by cold and dry winters with rainfall maxima occurring during summer in the upstream part where elevation ranges from 0 to 3905 m above sea level (a.s.l.). At higher altitudes, significant water resources accumulate during the winter season in the form of snowfall (Bertoldi et al., 2023), which are mobilised during spring/early summer determining a snowmelt-dominated hydrological regime (average discharge of 371 m³/s in June in Trento over the last 14 years; Floods.it, last access:10/03/2025). At lower altitudes, in the downstream plains, streamflow shows a pluvial regime with two peaks: one in autumn, mainly driven by intense cyclonic precipitation, and another one in spring/early summer, primarily due to snowmelt(average discharge of 310 m³/s in June and of 295 m³/s in November at Boara Pisani over the last 14 years; Ambiente Veneto, last access: 10/03/2025). As a mountain basin where glaciers and snowmelt mainly contribute to streamflow during late spring and summer, the system is highly sensitive to drought and temperature shifts: the combination of low snow cover in winter, increasing temperatures in early spring, early

snow melting altering the timing of runoff, and intense heatwaves in summer can lead to widespread impacts across multiple environmental and socio-economic sectors. Beyond the climatic drivers, competing water demands for irrigation, hydropower production, tourism, and ecosystem services add further complexity to the dynamics of water availability. These demands, particularly in downstream areas, can intensify impacts and give rise to resource conflicts (Chiogna et al., 2016).

# 3. Data and Methods

# 3.1 Data

To calculate drought and heatwave events separately we used the E-OBS v28.0 dataset (Cornes et al., 2018), which provides daily maximum temperature (TX) and total precipitation (TP) at a 0.1° spatial resolution from 1950 to 2023. Meteorological fields were analysed on a wider portion of the Eastern Alps (7°E-15°E, 44°N-49°N) centred on the Adige River basin. For comparison purposes, we also used the respective fields available in ERA5-Land reanalysis (Muñoz-Sabater et al., 2021), matching E-OBS resolution (0.1°, daily) to ensure consistency. In both cases, daily precipitation was aggregated to a monthly 145 scale when calculating the drought indicator. We acknowledge that the 0.1° resolution of E-OBS and ERA5-Land may introduce biases in representing local-scale variability as it has been already noticed in Bandhauer et al., (2021), particularly due to the complex orography of the Adige basin. However, to date no observational gridded dataset or reanalysis provides such a long and continuous record of daily temperature and precipitation for this region. The use of E-OBS therefore represents the best compromise between temporal coverage and spatial resolution, while ERA5-Land serves as a 150 complementary dataset to assess robustness. To characterise the synoptic-scale atmospheric circulation, we retrieved daily geopotential height at 500 hPa (Z500) from ERA5 (Hersbach et al., 2020) at its native 0.25° resolution, covering a broader domain (10°W-30°E, 30°N-60°N) over Western Europe.

To assess the performance of climate models in comparison to observations, we used the EURO-CORDEX ensemble (Jacob et al., 2014) under both the historical and RCP8.5 scenarios for the abovementioned daily variables TX, TP and Z500. The historical simulations covered the 1951–2005 period, while the RCP8.5 scenario extended from 2006 to 2100. To ensure the calculation of the circulation analogues described in section 3.3, we extended historical simulations to 2021 by merging them with the 2006–2021 segment from the RCP8.5 simulations as usually applied in similar studies (Zhang et al., 2023). According to Vautard et al. (2021), differences across various RCP scenarios are expected to be minimal in the first years of simulations. The RCP8.5 scenario also corresponds relatively well to observed emission trends (Zittis et al., 2019). The EURO-CORDEX ensemble consists of 25 members (Table A1), incorporating multiple regional climate models (RCMs) driven by different global climate models (GCMs), ensuring a robust representation of uncertainty and long-term climate trends over the Adige River basin.

To analyse observed changes in the Adige River streamflow, we used two datasets. First, we analysed daily discharge data from the Trento-Ponte San Lorenzo gauging station (Trento Province, Italy), obtained from the Alpine Drought Observatory (Alpine Drought Observatory, 2024), covering the period 1951–2021. Second, we used daily river discharge data from the

HERA (High-resolution Pan-European Hydrological Reanalysis) dataset (Tilloy et al., 2025), which provides a spatial reconstruction of river discharge with a 1.8-km resolution for the period 1951–2020. Noteworthy, HERA does not represent naturalized streamflow, as long-term changes in the dataset reflect not only climate variability but also the influence of human management. However, most of the largest dams and artificial lakes in the Adige basin (e.g., Santa Giustina, 182 Mm³, built in 1951; Resia, 120 Mm³, built in 1949; Stramentizzo, 11.5 Mm³, built in 1956) were already in place at the beginning of the study period. This suggests that the major hydropower infrastructure was largely stable during 1951–2020, even though operational strategies may have evolved over time. In any case, any direct link between climatic drivers and discharge trends should be interpreted with caution. In addition to the above, we incorporated daily snow depth data from three historical observatories located in the northwestern Adige catchment, spanning 1980–2018 and retrieved from Matiu et al. (2021), to investigate the impact of snow resources into streamflow dynamics. ERA5-Land monthly snow cover for the Adige basin from 1951 to 2020 for April, May, and June was also used to assess consistency with the river discharge patterns from HERA, as the latter was bias-corrected using ERA5-Land. Similarly, we used the global snow cover phenology dataset provided by Notarnicola (2024), which exploits the MODIS MODI0A1.061 product to derive yearly averaged snow cover area.

#### 3.2 Definition of compound drought and heatwave events

CDHW events were identified as the simultaneous occurrence of drought and heatwave conditions at grid cell level (Fig. 2). To account for drought conditions, daily precipitation records were aggregated to monthly totals and the Standardised Precipitation Index at a 6-month scale (SPI-6) was calculated (McKee et al., 1993). The SPI-6 was chosen as it effectively captures medium-term precipitation deficits relevant for hydrological and agricultural droughts (Oikonomou et al., 2020; Tsesmelis et al., 2019), making it particularly suitable for assessing impacts on river discharge, which responds to prolonged precipitation deficits rather than short-term fluctuations. A drought period was identified as a sequence of at least two consecutive months with negative SPI-6, starting with the first month with SPI-6 < -1. Heatwaves were characterised as periods of at least three consecutive days with daily TX above the 90<sup>th</sup> percentile for that specific calendar day (TX90), determined from a 31-day running mean, following a similar approach as in Russo et al. (2014). If two heatwave periods were separated by only one day with non-exceeding temperature, the two periods were counted as one single event (Lavaysse et al., 2018; Serrano-Notivoli et al., 2022), and the intermediate day was included as part of the heatwave. The reference period used for calculating SPI-6 and TX90 corresponded to the whole period spanned by E-OBS from 1950 to 2023. Regional CDHW events were identified when simultaneous hot and dry conditions affected at least 60 % of the Adige catchment area.

Figure 2. Workflow of detection of compound drought and heatwave (CDHW) event at single-cell level.

Detected CDHW events were characterized in terms of severity. At single-cell level, the daily CDHW severity (CDHW $sev_d$ , dimensionless) was calculated by combining drought intensity and heatwave intensity similarly as proposed by Tripathy & Mishra (2023) shown in eq. 1:

CDHWsev<sub>d</sub> =
$$(-1 \times SPI-6) \times \left(\frac{TX_d - TX_{p25}}{TX_{p75} - TX_{p25}}\right)$$
 (eq.1)

where SPI-6 is the grid cell value of SPI-6 in the month under CDHW conditions,  $TX_d$  is the daily maximum temperature, and  $TX_{pn}$  is the  $n^{th}$  percentile threshold for each calendar day after calculating a 31-day centred running mean. At the regional level, the CDHW severity was determined by averaging the severity values over all affected cells in the Adige River catchment. By multiplying the affected area fraction by the total CDHWsev, we generated a ranked list of events from 1950 to 2023, which allowed us to identify the May of 2022 as the most extreme CDHW event in the Adige River catchment over the last 15 years in terms of both area and severity; this event was consequently selected as a study case for the attribution analysis.

# 3.3 Flow analogues experiments

# 3.3.1 Observation analysis

235

240

To attribute the record-breaking compound dry-hot conditions of May 2022, we used circulation analogues, as described by Jézéquel et al. (2018). This technique aims to infer the behaviour of a target variable (e.g., temperature and precipitation) in a past or future climate (also referred to as the counterfactual world) as well as in the current conditions (factual world), based on the specific atmospheric circulation patterns of a given event (i.e., May 2022 CDHW). The key criterion for this inference is that the atmospheric circulation characteristics during factual and counterfactual periods must be analogue to those 215 observed during the target event. We set the factual and counterfactual periods to 1992-2021 and 1951-1980, respectively. Circulation-analogue days were defined from their root-mean-square differences (RMSD) with respect to each daily Z500 anomaly field at the time of the CDHW event. For each day of the considered event, the search of flow analogues was restricted to neighbouring months (i.e., for a CDHW in May, we considered April, May and June, AMJ), excluding the year of occurrence of the event. Note that Z500 was detrended prior to proceeding with the analogue search, ensuring that the 220 analogue selection was not biased by the global thermal expansion signal (Christidis and Stott, 2015). Analogue days were used to reconstruct the target field by randomly selecting one of the 20 best flow analogues for each day of the event. These 20 analogues represent approximately 0.7% of the total sample for each period, ensuring that only the synoptic patterns most similar to the event are retained while maintaining a representative subset for the analysis. This reconstruction was repeated 1000 times for the factual as well as the counterfactual periods. Following the above procedure, we first reconstructed the 225 probability distribution of TX, SPI-6, daily TP over the Eastern Alps domain. Anomalies in the factual and counterfactual worlds were defined for all reconstructed variables with respect to the period 1951–2021. For comparison, the distribution of anomalies was also randomly reconstructed but, in this case, not conditioned on the target event circulation, providing a control simulation that only reflects potential thermodynamic changes due to global warming (Barriopedro et al., 2020). Subsequently, to further assess the potential impacts on streamflow of changing hot-dry conditions, daily discharge 230 anomalies for the Adige basin were reconstructed by applying the same approach. The reconstruction of streamflow values in factual and counterfactual worlds allowed to understand the observed changes in potentially available water streamflow with implications for hydropower generation or agriculture.

Three metrics were defined to support the interpretation of the analogue-based attribution analysis and are subsequently compared between the counterfactual (1951–1980) and factual (1992–2021) periods. First, the analogue quality Q represents the average RMSD of every event day from its closest 20 analogues. Differences in Q between the counterfactual and factual periods provide insights into whether the atmosphere is transitioning toward states (analogues) that are more or less similar to the Z500 map associated with the target event. Additionally, changes in the distribution of Z500 for the 20 analogues reveal whether these atmospheric conditions are becoming more or less typical over time (Faranda et al., 2020). To evaluate this, the annual frequency of analogues, i.e., the number of analogue days per year, was investigated and a linear regression was fit to get the statistical significance of the frequency trend. Thirdly, the seasonality of analogues was analysed by

counting their monthly occurrence to identify shifts in analogues toward an earlier or later part of the season. Such shifts can affect thermodynamics significantly (Faranda et al., 2022); for example, a pattern causing positive temperature anomalies in late spring could have a stronger impact if it occurs in early summer, when average temperatures are already higher.

The analogue-based attribution procedure can be summarized as follows:

- 1. Define periods: select a counterfactual period (1951–1980) and a factual period (1992–2021).
  - 2. Identify analogues: for each day of the May 2022 CDHW event, find the 20 most similar days (flow analogues) from the reference periods, based on the RMSD of daily (detrended) Z500 anomaly fields. The search is restricted to neighbouring months (April–June), excluding the year of occurrence.
  - 3. Reconstruct events: stochastically reconstruct the event by randomly sampling one analogue per day, repeating this process 1000 times for each period.
  - 4. Generate distributions: compute probability distributions of maximum temperature (TX), precipitation (TP), SPI-6, and daily streamflow anomalies in the factual and counterfactual periods.
  - 5. Assess analogue properties: quantify changes in (i) analogue quality (RMSD), (ii) analogue frequency (number of occurrences per year), and (iii) analogue seasonality (monthly distribution), to assess shifts in atmospheric circulation associated with CDHW conditions.

To assess the role of interdecadal climate variability on the observed surface changes, we applied the methodology proposed by Faranda et al. (2022). Monthly teleconnection indices were obtained from NOAA/ERSSTv5 data via KNMI's Climate Explorer, specifically the Arctic Oscillation (AO) and the Atlantic Multidecadal Oscillation (AMO) (Trenberth & Shea, 2006). We performed a two-tailed Mann-Whitney U test at the 0.05 significance level in all cases to assess the significance of differences of AO and AMO between the factual and counterfactual distributions. A p-value less than 0.05 leads to the rejection of the null hypothesis (H=0) that the two samples come from identical distributions (Anderson, 1962). In other words, we cannot rule out the possibility that the thermodynamic or dynamical differences between the analogues are due in part to these natural modes of variability rather than to anthropogenic forcing. On the other hand, when H=1, we cannot reject the null hypothesis (H=0), suggesting that observed changes in analogues could be attributed to anthropogenic climate change rather than observed changes in the AO or AMO natural variability indices. It should be stressed, however, that comparing the distributions of teleconnection indices conditional on analogue circulation does not necessarily imply a causal relationship with the observed changes in circulation or impact variables. This limitation is particularly important for the AMO, given its long periodicity (Zampieri et al., 2017), which complicates the attribution of causal links between its variability and the surface anomalies analysed. Additionally, we recognise that other teleconnection indices not studied in this work could also play a role in the observed changes.

# 3.3.2 EURO-CORDEX analysis

The same approach described above for the detection and reconstruction of Z500 circulation analogues for TX and SPI-6 was applied to EURO-CORDEX simulations. As for observations, SPI-6 at a monthly scale was calculated from monthly TP, which was previously aggregated from daily to monthly.

The same periods used for the observation-based analysis (i.e., 1951-1980 and 1992-2021) were adopted, providing circulation-conditioned and unconditioned reconstructions for each period and model. To evaluate the performance of the models against the observations, we analysed their ability to reproduce the direction of change between periods (i.e., the sign of difference between 1992-2021 and 1951-1980) as well as the magnitude of the change.

# 4 Results

#### 4.1 Observed characteristics of the May 2022 CDHW event

In the late spring of 2022, a CDHW event hit the eastern Alps, with the most severe conditions observed in the southern half of the region, affecting over 80% of the Adige catchment (Fig. 3a). The event, which spanned from May 10 to May 28, was triggered by unprecedented drought conditions, with more than 50% of the catchment experiencing long-lasting precipitation deficits accumulated over the previous 6–12 months (Fig. A1a–e). In addition to the drought conditions, temperature anomalies recorded during this period were exceptionally high, with record-breaking values in some areas of the Adige basin, particularly in its upper and higher-elevation regions (Fig. A1f–j). The anomalous thermodynamic conditions of this event were partly driven by a strong geopotential height anomaly at 500 hPa (Figure 3b), reaching up to 120 m above average over the Alps. This atmospheric pattern contributed to the persistence of hot conditions across central Europe, enhancing the intensity of the heatwave. In terms of event severity, the compound extremeness of the event ranked as the fifth most extreme since 1950 and was the most severe CDHW event observed in the past 15 years (Fig. 3c). As a result, 2022 emerged as the second most extreme year on record after 2003 in terms of annual CDHW severity.

Figure 3. (a) Total severity values derived from the CDHWsev indicator and (b) 500-hPa geopotential height anomalies for the event from 10 to 28 May of 2022. c) Severity-area weighted ranked events for the Adige catchment from 1950 to 2023. The upper-right inset plot in c) shows the annual total severity-area weighted. Event and year 2022 are pointed out in red.

# 4.2 Observed changes in the meteorological drivers exacerbating CDHW severity

To investigate the role of global warming in enhancing the intensity of the May 10-28 event in 2022, we reconstructed the atmospheric configurations leading up to the study case using a stochastic simulation based on random sampling from the closest analogues (Fig. 4). First, we reconstructed the observed anomalies averaged over the entire event for both the

counterfactual (1951-1980) and factual (1992-2021) periods. This includes the Z500 anomalies (Fig. 4a-c) as synoptic variables, and SPI-6 (Fig. 4d-f), TX (Fig.4g-i), and TP (Fig. 4j-l) as surface variables.

The reconstructed fields revealed significant changes in the synoptic conditions associated with the event. In particular, Z500 exhibited a pronounced increase over central Europe in the factual period (Fig. 4c) indicating a strengthening of the large-scale atmospheric configuration that triggered the May 2022 heatwave. This suggests that analogous synoptic situations have become more intense in recent decades compared to the counterfactual period (1951–1980). This increase in geopotential height could be explained by the warming of the air column.

In terms of surface variables, the SPI-6 revealed a clear drying pattern across almost the entire Adige catchment (Figure 4f). Notably, the southern part of the catchment shifted from a wetter-than-normal state (Figure 4d) to a drier-than-normal state (Figure 4e), a transition that is consistent with the drying pattern observed across the southern and south-eastern regions of the extended alpine domain. In contrast, the rest of the study area did not exhibit substantial changes in moisture conditions. Only the north-eastern end of the basin showed an opposite pattern, going from a drier to a wetter-than-normal state. As regards mean temperature anomalies over the entire event, the regions exhibiting the greatest warming are located in the southern and south-eastern part of the Adige basin, where the drying process shown by SPI-6 is also more pronounced (Fig. 4g-i). This consistency between temperature and SPI-6 pattern changes might suggest possible positive feedback between dry conditions and high temperatures (Zhang et al., 2023; Lemus-Canovas et al. 2025).

Temperature patterns are also consistent with the total observed precipitation changes when the circulation is conditioned to the target event (Fig. 4j-l). The reconstructed precipitation in factual and counterfactual periods shows higher amounts in the area surrounding the northern part of the Adige basin. This can suggest that the relatively wetter conditions in this area, despite slight variations between the two periods, helped to limit the maximum temperature increases between +1°C and +1.5°C with respect to 1951-1980. In contrast, in southeastern part of the basin and close surroundings, precipitation for an event like that of May 2022 decreased by almost 10-15 mm in 1992-2021 with respect to the past period. This precipitation reduction roughly coincides with a temperature increase beyond that expected from global warming, quantified locally at +4°C, which might be favoured by more incoming radiation due to fewer precipitation days.

Figure 4. Analogue reconstruction of the May 2022 event using anomalies of key climate variables for past (1951–1980) and present (1992–2021) periods. Panels (a-c) depict geopotential height at 500 hPa (Z500), while panels (d-f) show Standardized

Precipitation Index at a 6-month scale (SPI-6). Panels (g-i) present daily maximum temperature (TX) anomalies, and panels (j-l) illustrate daily total precipitation (TP) anomalies. The first two columns show anomalies for the past and present periods, computed relative to the 1951–2021 baseline. The third column displays the difference between present (1992–2021) and past (1951–1980) periods for each variable.

To summarise the above observed surface changes, we provided a flow conditioned and a random unconditioned reconstruction over the Adige basin (Fig. 5). In general, statistically significant shifts in hydroclimatic conditions were observed between the periods 1951–1980 and 1992–2021. TX anomaly showed a clear increase in the more recent period (Fig. 5a). Flow-conditioned reconstructions exhibited markedly higher values, reflecting pronounced regional warming, while unconditioned reconstructions showed greater variability but remained consistent with the overall warming trend. Furthermore, the SPI-6 shows a significant decrease in the period 1992-2021 (Fig. 5b). Unconditional reconstructions show a drying trend, transitioning from slightly positive to negative SPI-6 values (as median). In the case of TP, the difference in the median between the two periods is less clear (Fig. 5c). However, for the more recent period, both the flow-conditioned and unconditional reconstructions show reduced variability, as well as a statistically significant tendency towards a decrease in daily precipitation..

Figure 5. Comparison of hydrometeorological variables in the Adige River basin for the periods 1951–1980 and 1992-2021. Boxplots of (a) daily maximum temperature anomalies (TX), (b) SPI-6, and (c) daily precipitation (TP), with flow-conditioned reconstructions (colored boxes) and unconditioned reconstructions (gray boxes). To assess the statistical significance of differences between the two periods, we applied the Mann-Whitney U test at 0.05 (\*) significance level. The absence of statistical significance is reported as "ns".

We also conducted a series of tests to evaluate the influence of natural variability on the observed increases in temperature and drought conditions (Fig. 6). Our results indicated that circulation structures, such as the one in May 2022, have become

more common compared to the past climate (Fig. 6a), a change linked to increased geopotential heights in Z500. However, while these patterns (i.e., analogue days) are more frequent, the upward trend remains statistically insignificant (Fig. 6b). We further examined seasonal shifts in analogue occurrences between past and present climates (Fig. 6c). Notably, there is a slight shift from late spring to early spring, with events like that of May 2022 occurring slightly earlier than in previous decades. This earlier occurrence could theoretically result in lower temperatures associated with the event in the latter period compared to the former. Nevertheless, as illustrated in Fig. 6a, a clear increase in temperature persists, driven primarily by global warming rather than the accounted minor shifts in seasonality. However, in the following section, we further investigate these temporal shifts in relation to potential changes in streamflow dynamics. Finally, we examined the potential role of large-scale teleconnection patterns, such as AMO and AO, in modulating these circulation changes and their effects on surface variables (Fig. 6d-e). Our analysis shows that neither AMO (Fig. 6d) nor AO (Fig. 6e) exhibit statistically significant changes between the two periods, indicating that the considered teleconnections do not play a significant role in amplifying the intensity of recent events like May 2022. Thus, our findings suggest that recent changes in circulation and associated surface impacts are expected to be driven more by regional and global warming trends than by teleconnection influences.

Figure 6. Statistical analysis of the analogue days to 10-28 May 2022. Panel (a) shows the Root Mean Square Deviation (RMSD) of analogues between 1951–1980 (blue) and 1992–2021 (red). Panel (b) illustrates the annual frequency of analogue days, with a

linear regression fit indicating no statistically significant trend. Panel (c) presents the seasonality of analogue occurrences based on monthly frequencies per period. Finally, panels (d) and (e) display the distributions of the Atlantic Multidecadal Oscillation (AMO) and Arctic Oscillation (AO) indices for the two periods. All statistical comparisons are based on the Mann-Whitney U test at 0.05 (\*) significance level.

#### 4.3 Seasonal shifts in streamflow and the impact of early snowmelt

In addition to analysing observed changes in meteorological variables, we investigated hydrological changes by examining streamflow contributions of the Adige River at Trento in relation to atmospheric circulation analogues. The results indicate that streamflow variations depend on the month in which analogues are taken, following a pattern consistent with the snowmelt-dominated pluvial regime characteristic of the Adige River in Trento (Mallucci et al., 2019; Chiogna et al., 2016). This regime is primarily controlled by spring snowmelt and autumn precipitation events. During the 1951–1980 period, we observed a progressive increase in streamflow as analogues shift from April to June (Fig. 7a). However, in the more recent period (1992–2021), this pattern changes significantly. Specifically, in April and May, Adige streamflow slightly increased (not statistically significant) compared to the 1951–1980 period. However, in June, streamflow exhibits a statistically significant decline in the recent period compared to the past (Fig. 7a). These monthly differences translate into a statistically significant overall reduction in streamflow for all analogues in April, May, and June (Fig. 7b). This overall streamflow reduction cannot be attributed to changes in the timing of the analogues, as the differences between the two periods across the months are almost negligible (Fig. 7c and Fig. 7d).

Figure 7. (a) Boxplots of analogues streamflow in the Adige River at Trento for April, May, and June in the periods 1951-1980 (blue) and 1992-2021 (red). (b) Overall streamflow in the two periods across all analogues, with statistical significance assessed using the Mann-Whitney U test at 0.05 (\*) significance level. (c) Monthly relative weights of the number of analogue days. (d) Percentage change in analogue days distribution between the two periods for each month from April to June. (e) Observed snow depth 30-days averaged climatology (solid lines) and their respective day of the year trends over 1981-2018 period (dotted lines) at

three high-altitude stations in the upper Adige catchment (circles on the inset map). (f) The same as (e) but for streamflow at Trento gauge station.

Notably, given the observed changes in the river regime, the timing of a CDHW event can lead to substantially different hydrological impacts. This highlights the importance of understanding the drivers of this shift in the Adige River regime. To investigate this aspect, we analysed the temporal evolution of snow depth at three in-situ measurement points in the upper Adige catchment, all located above 1800 m (Fig. 7e). The results show that in early spring -as temperatures rise and the snowpack begins to decline rapidly- snow depth decreased by approximately 30-40 cm per 30 years based on long-term 400 trends in mid-April (Fig. 7e, dotted lines), with values of -36 cm at Roia di Fuori (95% CI: -60.1 to -12.8 cm; Figure A2a), -28 cm at Fontana Bianca (-52.1 to -1 cm), and -6 cm at Diga di Gioveretto (-26.7 to 22.3 cm). Such results are also in line with a statistically significant decrease in the snow cover fraction derived from satellite data above 1500 m (Fig. A3). This translates into an earlier onset of snowmelt, which in turn led to a trend increase in Adige streamflow of up to 38 m<sup>3</sup>/s in late April to early May over the same period (Fig. 7f, 95% CI: -19.1 to 81.3 m<sup>3</sup>/s; Figure A2b). In addition, this 405 initial upward trend was followed by a subsequent streamflow decline up to -50-60 m<sup>3</sup>/s (30-year trend) by mid-June (95%) CI: -113.1 to 28.3 m<sup>3</sup>/s). This suggests that premature snowmelt can shift the spring peak streamflow earlier in the season, while simultaneously leading to an earlier transition into summer low-streamflow conditions. As a consequence, a CDHW event in April or early May could now encounter more water in the Adige River than in the past, while a CDHW event in June could coincide with lower streamflow than in the past thus exacerbating the effects of evapotranspiration and water 410 demand which are generally higher due to the warmer conditions of the summer season.

To assess these hydrological changes across the entire Adige basin, we reconstructed the relative changes in river discharge, using streamflow data from the HERA dataset (Figure 8). In April, we observed increased discharge across most of the basin, which can be related to the acceleration of snowmelt processes in recent decades. This pattern in fact aligns with the earlier onset of snowmelt highlighted in Figure 7e, suggesting that warming not only anticipated the timing of snowmelt but also intensified meltwater contributions during this month. However, the increased discharge is not statistically significant given the low number of analogues selected in April (Fig. 7c).

Figure 8. Relative changes in river discharge (%) in the Adige River basin between the periods 1992–2021 and 1951–1980 for April, May, and June. Black dots indicate statistically significant differences at the 95% confidence level. ERA5-Land pixels with at least 95% snow cover during the month, averaged over the 1951–2020 period, are highlighted in magenta for April and May. For June, the threshold was lowered to 70% to better capture snowmelt buffering areas.

Moving into May, the discharge pattern shifts. Positive relative anomalies are primarily confined to the upper catchment areas, coinciding with the highest elevations where snow reserves persist longer, and melting is further exacerbated by recent warming (Fig A3). However, in the mid and lower catchments, we observe a transition towards relative discharge deficits. This reflects the earlier depletion of snow reserves due to accelerated melting in April as noted in Fig. A3, leaving less water available for runoff as the season progresses. By June, we observed a localised relative increase or sustainment in discharge at the northern end of the basin, mainly due to enhanced snowmelt at higher elevations —where snow persists later into the season (magenta points; also aligned with MODIS snow cover product in Fig A3a) — in response to ongoing regional warming. However, across most of the basin, there are pronounced relative decreases in discharge, with anomalies ranging from –80% to –20%, consistent with the patterns observed at the Trento gauge station (Figure 7f).

#### 4.4 Do EURO-CORDEX simulations reflect the observed changes for events like May 2022?

After considering the observed hydroclimatic changes for an event like May 2022, we aimed to assess the ability of 25 EURO-CORDEX simulations to capture the same temporal patterns as observed for Z500, SPI-6 and TX. We repeated the same methodological process of event reconstruction over the Adige basin similar to that shown in Figure 4, but in this case applied to the fields directly derived from each of the GCM-RCM simulations. We approached the evaluation based on two

main aspects: 1) number of models that reproduce the same sign of change between periods as in the observations; 2) number of models that reproduce a similar magnitude of change as the observations.

For temperature circulation-conditioned reconstructions (Fig. 9a,b), about 14 (56%) models are able to capture the positive sign of the change as in the observations, but strongly underestimate it (13 out of 14 show an increase less than +0.5°C while E-OBS and ERA5 Land estimated a general increase over +2°C and +1°C, respectively; Fig A4). In contrast, all 25 (100%) models reproduce the same sign of the change when they are unconditioned to the circulation of the event (Fig. 9c, d). However, the models still underestimate the magnitude of the change: only 12 (48%) of them show an increase of more than +1°C. The underestimation in TX corresponded with an underestimation of the expected increase in Z500 values in the more recent period, even when considering the 5 models closer to observations (Fig. A5 and Fig. A6). In other words, such underestimation of near surface temperature can be also transferred to the rest of the air column: when analyzing heatwave events analogue to May 2022 (Fig. A7), only 6 (24%) models reproduce the same sign of change jointly in Z500 and TX. Additionally, they also fail to reproduce the trend in the annual frequency of analogues shown by ERA5 (Fig. A8)

Figure 9. Comparison of daily maximum temperature (TX) reconstructions for the periods 1951–1980 and 1992–2021 using flow-conditioned (panels a-b) and non-conditioned (panels c-d) approaches. Panels (a) and (c) display boxplots of temperature anomalies (°C) for individual models and the E-OBS dataset (grey shaded boxplot), with blue and red boxes representing the 1951–1980 and 1992–2021 periods, respectively. Panels (b) and (d) show the differences in temperature anomalies between the two periods ([1992–2021] - [1951–1980]) for each model, highlighting changes across the models. Pink markers in panels (b) and (d) indicate models with same sign of change as in observations (E-OBS, cyan marker), while the red dashed lines indicate the median value for each E-OBS boxplot in (b) and (d). The complete names of the models are detailed in table A1.

Regarding drought conditions, EURO-CORDEX simulations are not able to reproduce the same drying processes as observed in E-OBS for SPI-6, either conditioned (5 models, 20%) or unconditioned (4 models, 16%) to the circulation of the May 2022 event (Fig. 10). In any case, we would like to acknowledge that for SPI-6 the unconditioned and conditioned reconstructions perform quite similarly, because the drought is not directly related to the circulation of the May event. In

other words, the cumulative drought conditions are not caused by the May event itself, but providing these analyses are still useful to know if we can expect drought conditions to concur with a heatwave event.

# 465 Figure 10. As in Figure 9 but for monthly SPI-6.

In conclusion, there are not well-performing model simulations in both variables when reconstructing a situation like in May 2022. We found only three GCM-RCMs are able to represent the observed signal of change in both variables, but not in terms of magnitude, especially after considering their temperature increase (

Figure A1. Maps of Standardized Precipitation Index (SPI) values at different time scales (SPI-3, SPI-6, SPI-12, SPI-24) relative to the 1950–2021 baseline, during May 2022 (panels a–d). Panel e displays the 1950-2021 time series of the area fraction of the Adige River catchment under varying drought severities. Panels f–i show daily maximum temperature anomalies (TX, relative to 1950–2021) averaged over 7-, 15-, 31-, and 61-day periods during May 2022. Magenta dots mark record-breaking TX values, while grey dots indicate the second-highest values for each grid cell. Panel j illustrates the temporal evolution of the record-breaking area (km²) during May 2022 within the Adige River catchment, assessed over moving windows of 1 to 61 days.

Other appendix figures:

Figure A2. (a) Snow depth at three historical observatories in the northwestern Adige catchment (Diga di Gioveretto, Fontana Bianca, and Roia di Fuori) with 30-year day-of-year trends and their 95% confidence intervals over 1981–2018. (b) Same as (a) but for Adige River discharge at the Trento gauge station.

Figure A3. a) Average snow cover percentage for the period 2000–2023 by elevation band in the Adige catchment, derived from MODIS data. (b) Decadal trends in snow cover percentage over the same elevation bands. Cells highlighted with a black border denote trends that are statistically significant at the 95% confidence level.

Figure A4. Spatial comparison of flow conditioned daily maximum temperature anomalies (TX) reconstructed for the periods 1951–1980 and 1992–2021, and the differences between these periods ([1992–2021] - [1951–1980]), based on three datasets: E-OBS (top row), ERA5-Land (middle row), and EURO-CORDEX ensemble mean (bottom row). The first two columns represent temperature anomalies for the two periods, while the third column shows the difference in anomalies.

Figure A5. Geopotential height flow conditioned anomalies at 500 hPa (Z500) for the periods 1951–1980 (a, d) and 1992–2021 (b, e), and the differences between the two periods ([1992–2021] - [1951–1980], c, f). Panels (a–c) show ERA5-based reconstructions, while panels (d–f) display results from the top 5 EURO-CORDEX models, selected based on their analogue quality (Fig. S4).

 $Figure\ A6.\ Sames\ as\ Figure\ A3,\ but\ for\ TX.\ The\ observed\ reconstruction\ is\ derived\ from\ E-OBS.$ 

Figure A7. Comparison of analogue quality for geopotential height at 500 hPa (Z500) reconstructions for the periods 1951–1980 and 1992–2021. a) displays boxplots of Root Mean Square Deviation (RMSD) values for the entire 1951–2021 period across individual models and the ERA5 dataset (grey shaded boxplot); b) compares RMSD values for the periods 1951–1980 (blue) and 1992–2021 (red), showing analogue quality across models; c)illustrates the differences in RMSD between the two periods ([1992–2021] - [1951–1980]) for each model, highlighting changes in analogue quality. Pink markers in the bottom panel indicate models with same sign of change as in observations. The complete names of the models are detailed in table A1.

Figure A8. Annual frequency of analogue days based on geopotential height at 500 hPa (Z500) from 1950 to 2021, comparing ERA5 (black line) and the EURO-CORDEX multi-model ensemble (gray line with shading representing the inter-model spread across 25 EURO-CORDEX models based on 25th and 75th percentiles, respectively). Linear regression fits are shown for both ERA5 (dashed black line) and the multi-model ensemble (dashed gray line).

Figure A9. Averaged Z500 over the May 2003 and 2022 episodes.

Figure A10. Number of analogues of the May 2022 event detected by year.

Table A1. List of EURO-CORDEX simulations used in this study.

| Sim | GCM                   | RCM                          | GCM-RCM abbreviation |
|-----|-----------------------|------------------------------|----------------------|
| 1   | CCCma-CanESM2         | GERICS-REMO2015              | CanESM2_REMO2015     |
| 2   | CCCma-CanESM2         | CLMcom-CCLM4-8-17            | CanESM2_CCLM         |
| 3   | CNRM-CERFACS-CNRM-CM5 | CNRM-ALADIN63                | CNRM-CM5_ALADIN      |
| 4   | CNRM-CERFACS-CNRM-CM5 | DMI-HIRHAM5                  | CNRM-CM5_HIRHAM5     |
| 5   | CNRM-CERFACS-CNRM-CM5 | GERICS-REMO2015              | CNRM-CM5_REMO2015    |
| 6   | CNRM-CERFACS-CNRM-CM5 | KNMI-RACMO22E                | CNRM-CM5_RACMO22E    |
| 7   | ICHEC-EC-EARTH        | DMI-HIRHAM5                  | EC-EARTH_HIRHAM5     |
| 8   | IPSL-IPSL-CM5A-MR     | DMI-HIRHAM5                  | IPSL-CM5A_HIRHAM5    |
| 9   | IPSL-IPSL-CM5A-MR     | KNMI-RACMO22E                | IPSL-CM5A_RACMO22E   |
| 10  | IPSL-IPSL-CM5A-MR     | GERICS-REMO2015              | IPSL-CM5A_REMO2015   |
| 11  | MIROC-MIROC5          | GERICS-REMO2015              | MIROC5_REMO2015      |
| 12  | MIROC-MIROC5          | CLMcom-CCLM4-8-17            | MIROC5_CCLM          |
| 13  | MOHC-HadGEM2-ES       | DMI-HIRHAM5                  | HadGEM2_HIRHAM5      |
| 14  | MOHC-HadGEM2-ES       | MOHC-HadREM3-GA7-05          | HadGEM2_HadREM3      |
| 15  | MOHC-HadGEM2-ES       | GERICS-REMO2015              | HadGEM2_REMO2015     |
| 16  | MOHC-HadGEM2-ES       | CLMcom-ETH-COSMO-crCLIM-v1-1 | HadGEM2_COSMO        |
| 17  | MOHC-HadGEM2-ES       | CNRM-ALADIN63                | HadGEM2_ALADIN       |
| 18  | MPI-M-MPI-ESM-LR      | DMI-HIRHAM5                  | MPI-ESM_HIRHAM5      |
| 19  | MPI-M-MPI-ESM-LR      | CNRM-ALADIN63                | MPI-ESM_ALADIN       |
| 20  | MPI-M-MPI-ESM-LR      | KNMI-RACMO22E                | MPI-ESM_RACMO22E     |
| 21  | MPI-M-MPI-ESM-LR      | MOHC-HadREM3-GA7-05          | MPI-ESM_HadREM3      |
| 22  | NCC-NorESM1-M         | MOHC-HadREM3-GA7-05          | NorESM1_HadREM3      |
| 23  | NCC-NorESM1-M         | DMI-HIRHAM5                  | NorESM1_HIRHAM5      |
| 24  | NCC-NorESM1-M         | CNRM-ALADIN63                | NorESM1_ALADIN       |
| 25  | NCC-NorESM1-M         | GERICS-REMO2015              | NorESM1_REMO2015     |

# Data availability

The compound drought and heatwave (CDHW) indicator for the Adige basin is available at Zenodo (Maines et al., 2025): <a href="https://zenodo.org/records/14859795">https://zenodo.org/records/14859795</a>

All datasets used in this study are publicly accessible through the following sources:

- E-OBS was used to calculate the CDHW indicators and to assess observed changes in TX and SPI-6. The data can be accessed at the E-OBS data portal: https://surfobs.climate.copernicus.eu/dataaccess/access\_eobs.php#datafiles
- ERA5 was used to derive circulation analogues. The dataset is available from the Copernicus Climate Data Store: https://doi.org/10.24381/cds.adbb2d47
  - EURO-CORDEX simulations were obtained from the Copernicus Climate Data Store.
  - HERA reanalysis was employed to estimate observed changes in streamflow levels across the Adige basin. The dataset can be accessed at the JRC Data Catalogue (Tilloy et al., 2025): https://data.jrc.ec.europa.eu/dataset/a605a675-9444-4017-8b34-d66be5b18c95
- The time series of river discharge for the Adige river at its passage through Trento-Ponte San Lorenzo has been retrieved from the Alpine Drought Observatory portal: https://edp-portal.eurac.edu/cdb doc/ado/ado/
  - Snow depth time series used in this work were obtained from Zenodo (Matiu et al., 2021): https://zenodo.org/records/5109574
  - The snow cover derived from MODIS used in this work was obtained from Zenodo (Notarnicola, 2024): https://zenodo.org/records/11181638

# Code availability

The flow analogues approach can be reproduced using the climattR package: https://github.com/lemuscanovas/climattR

#### **Author contributions**

M.L.-C. conceptualized the study, conducted all analyses, and wrote the first manuscript draft. A.C. and E.M. contributed to the conceptualization, definition, and computation of the CDHW indicator, as well as to the writing and reviewing of the manuscript. S.T. secured funding and contributed to the writing and revision of the manuscript. M.P. contributed to the manuscript review.

# Acknowledgements

This research was supported by the European Space Agency (ESA) under the EO4MULTIHA project (2023–2025), contract number 4000141754/23/I-DT. M.L.-C. was supported by a postdoctoral contract from the program named "Programa de axudas de apoio á etapa inicial de formación posdoutoral (2022)" funded by Xunta de Galicia (Government of Galicia, Spain). Reference number: ED481B-2022-055.

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
