# Peer review of "More intense heatwaves under drier conditions: a compound event analysis in the Adige River basin (Eastern Italian Alps)"

_EGUsphere, 2025_

## Referee Comment (RC1)

**Review of the research article titled "More intense heatwaves under drier conditions: a compound event analysis in the Adige River basin (Eastern Italian Alps)" and submitted to Hydrology and Earth System Science.**

**General Comments**

This article presents a study on the impact of climate change on the intensity of compound drought-heatwave (CDHW) events affecting the Adige River Basin, a major basin in the Italian Alps. The authors use E-OBS and ERA5 data to characterize past meteorological conditions, plus two datasets from the Alpine Drought Observatory and the High-Resolution Pan-European Reanalysis to analyse the streamflow of the Adige. Then, they consider an ensemble of 25 EURO-CORDEX RCMs to assess the performance of climate models in reproducing historical trends in CDWH.

The CDWH are defined using daily maximum temperature (TX) and total precipitation (TP) transformed to obtain the SPI-6 index. After detecting all CDWH events since 1950 in ERA5, May 2022 is chosen as a case study, being the second most intense event after 2003, and the most intense in the last 15 years. To study trends in temperature and precipitation associated with this type of event, flow analogs based on 500 hPa geopotential height anomalies (Z500) are adopted. This allows to condition the analysis of the relevant meteorological parameters to the synoptic configuration that produced the specific event, and is a popular strategy to study (compound) meteorological and climatological extremes. For comparison, unconditional trends are also considered.

As it is often done in this type of study, the 1950-2022 reanalysis period was divided into a counterfactual (1951-1980) and a factual (1992-2021) periods, and changes in Z500, SPI-6, TX, TP are quantified between the two periods. Changes in quality of the analogs, annual analog frequency, seasonal distribution are also assessed, together with a simple evaluation of the influence from two large-scale natural variability modes. The analysis is then repeated using EURO-CORDEX simulations, extending the historical period to 2021 using RCP8.5 simulations.

The research question is interesting and coherent with current concerns about the impact of anthropogenic climate change on extreme events, and is well within the scope of HESS. The paper reads very well, all parts are well structured and clearly presented, and the title clearly states the scope of the study. Conclusions are clear and non trivial, and sustained by well described results, obtained with overall robust methodology. The results about the EURO-CORDEX performance (or lack thereof) are particularly interesting and useful, since great attention is being given to the future evolution of extreme events under global warming scenarios, and understanding how reliable models are for specific types of event is important. Also, figures are overall easy to read and well support the interpretation of results.

I found the paper to be overall almost ready for publication, however I would like to point to some minor issues that the authors should be able to address quite easily, detailed in the next section.

Very minor linguistic imprecisions are present here and there, I will only point out one or two that might affect text clarity, the rest should be addressed by the editors.

**Specific Comments**

**Statistical testing**

This comment is general and concerns the execution and presentation of statistical tests used to assess differences in the distribution of several variables.

Overall, I advise against reporting significance at different levels using starts. Although this is a common practice, classic statistical inference is structured around the idea of controlling the probability of type 1 error - the level of the test - and the latter should be specified before running the experiment. I suggest either fixing the level at alpha = 0.05 and reporting only significance at this level, or dropping the significance reporting and directly showing the p-value as in Figure 6. This particularly concerns Fig. 5 and 7a.

All differences throughout the paper are tested using a two-sample Cramer von Mises test. This test compares the integral difference between the empirical probability distribution functions of the two samples, and it is therefore adequate to assess overall changes in the distribution. However, most of these tests aim at testing a shift in the distribution. While in cases such as in Figure 5 the resulting significant differences are clearly due to a shift, in other occasions this is not the case. In particular, the analog quality in Fig. 6a results significantly different between the two periods, however this difference is very likely due to a change in variability, not a shift; the authors comment on this result stating that the circulation becomes more common, but I would say that this is not the case. To solve this, I suggest to use a different non-parametric test focused on the central tendency of the distribution, and not on the comparison of the entire distribution functions. A good alternative could be the Mann-Whitney U test.

**Section 3.3.1**

While there are several ways to use analogs to perform detection or attribution, the authors here follow the method proposed by Jezequel et al. (2018). In this case, analogs are used to perform a simple stochastic weather generation of events analogs to the May 2022 CDHW, which is not the only possible way to use analogs in general. The methodology is very briefly summarized at lines 186-190. I would suggest adding a few words to explain the technique, possibly using a bullet-point list of the steps, since this explanation could be a little hard to grasp for someone who is not used to the technique.

At the end of the section, the evaluation of natural variability is explained, and some limitations are stated. This method -comparing the distributions of the indices in correspondence of the analogs in the two periods - is currently being used not only in other studies, but also in rapid attribution projects (e.g. ClimaMeter). I would say that the main limitation is that, even if significant changes are (or not) observed, it is very difficult to make any statement about the causal relationship or even the correlation between this difference and those observed in the meteorological variables, especially for the AMO, that has a very

long typical period. I suggest explicitly cautioning the reader about the fact that looking at teleconnection indices conditional to the circulation is not the same as looking at changes in circulation or impact variables caused by these teleconnections.

**Section 4.2**

Line 255, Figure 4: a clear trend in Z500 is found. However, as also stated later in the article, thermal expansion due to global warming directly causes an increase in geopotential heights. This does not equate observing a change in the circulation, since this is a global trend; Z500 should be detrended before performing this analysis, with previous studies suggesting to use a cubic rather than linear trend specification.

**Minor comments**

Line 20: the factual period is stated to be 1991-2020, while in the rest of the paper is 1992-2021.

Lines 112-113: "*...intense precipitations in autumn are mainly driven by cyclonic storms, and in spring/summer are due to snow melt processes, leading to a pluvial regime with two streamflow peaks…*" I might be missing something as I am not particularly expert in hydrology, but this sentence seems to state that snow melting is the cause of intense precipitation in the warm season. The authors probably mean that the two streamflow peaks are respectively due to strong cyclonic precipitation in autumn and snow melt processes in spring/summer, this sentence should be corrected.

Line 247: "*...we randomly reconstructed the atmospheric configurations […] based on the closest 20 analogs…*" I would rather say "we reconstruct the atmospheric configurations [...] using a stochastic simulation based on random sampling from the closest analogs". While there is a random component, most of the work is done by the analogs selection which is deterministic.

---

## Referee Comment (RC2)

I have carefully examined the manuscript titled "More intense heatwaves under drier conditions: a compound event analysis in the Adige River basin (Eastern Italian Alps)" (egusphere-2025-1347-2), submitted by Marc Lemus-Canovas, Alice Crespi, Elena Maines, Stefano Terzi, and Massimiliano Pittore from the Center for Climate Change and Transformation at Eurac Research, Bolzano-Bozen, Italy. This study investigates the increasing intensity and impacts of compound drought and heatwave (CDHW) events in the Adige River basin, with a particular focus on the significant event of May 2022. The authors employ a ranking of CDHW events from 1950 to 2023 using E-OBS data, a flow-analogue attribution approach with ERA5 geopotential height data, and an evaluation of EURO-CORDEX simulations to assess historical changes and future projections. Below, I provide my critical comments and recommendations to enhance the scientific rigor, clarity, and contribution of this work for publication in HESS journal.

1) The abstract and introduction effectively outline the problem, highlight the 2022 CDHW event, and introduce the attribution methodology, making it accessible to a broad audience. The transition from the abstract to the introduction lacks fluidity. The abstract references a ranking of 119 events and the selection of the 2022 event but omits details on the ranking methodology or the rationale for the 1950-2023 timeframe, leaving a disjointed narrative.

2) The abstract briefly mentions the use of E-OBS data, a composite index, and the flow-analogue attribution approach with ERA5 data, but it lacks specificity. For instance, what components constitute the composite index (e.g., temperature, precipitation, spatial extent)? How was the 1-4°C increase in heatwave intensity determined? Providing a brief methodological outline would enhance transparency and allow readers to assess the robustness of the findings upfront.

3) The discussion of atmospheric circulation patterns (e.g., subtropical ridge, warm air from northern Africa) is informative but lacks quantification. Terms like "prolonged periods" and "pronounced precipitation deficits" are vague without supporting data or references to specific magnitudes observed in the Adige basin.

4) The introduction highlights the scarcity of attribution studies at the catchment scale and the unexplored performance of EURO-CORDEX models, which is a strong motivation. However, it does not preview the specific attribution method (flow-analogue approach) or the limitations of EURO-CORDEX simulations (e.g., spatial resolution, parameterization), which are critical for setting expectations.

5) The use of E-OBS and ERA5 data is mentioned, but their resolution and potential biases (e.g., E-OBS's coarse grid in mountainous areas) are not addressed. This is particularly relevant given the Adige basin's complex topography.

6) The introduction cites several studies (e.g., Viviroli et al., 2007; Hao et al., 2022) to establish the importance of the Alpine region and compound extremes but lacks a critical synthesis. For example, it does not address whether previous studies have underestimated snow dynamics or elevation-dependent warming in the Alps, which are highlighted as unique challenges. I strongly recommend considering these two studies: Assimilation of sentinel-based leaf area index for modeling surface-ground water interactions in irrigation districts; Elevation dependent change in ERA5 precipitation and its extremes.

7) The abstract's note that over half of the EURO-CORDEX models failed to reproduce observed changes suggests potential issues with model selection or validation. The introduction does not foreshadow this, which could undermine confidence in the projections.

8) The streamflow story leans on one gauge (Trento) plus HERA; June reductions are attributed largely to earlier snowmelt. Please (i) discuss/quantify confounding from irrigation/hydropower

operations (not just note restrictions), (ii) report whether HERA is "naturalized" or includes management, and (iii) add a simple basin water-balance perspective (snow cover, PET/ET0, soil moisture) to separate supply vs. demand effects.

9) You show earlier snowmelt and an April/May discharge bump followed by June deficits. Consider cross-checking with independent snow data (in situ SWE, satellite snow extent) and add confidence intervals for the reported "30–40 cm per 30y" and "±40–60 m³/s" trends.

10) Beyond sign/magnitude counts, include formal skill scores (bias, RMSE, correlation, CRPS) for both conditioned and unconditioned reconstructions, and try simple emergent-constraint or performance-based weighting to see if an informed subset reduces the underestimation. Clarify implications of stitching historical with RCP8.5 to 2021.

11) Where you claim significant changes, consistently show effect sizes with CIs. You use Cramér–von Mises for some tests; extend uncertainty quantification to the event ranking, severity composites, and the discharge change maps (e.g., bootstrap over analogues and spatial blocks).

12) Since analogues are conditioned on one pattern, stress-test conclusions by repeating the pipeline for another major CDHW (e.g., 2003/2018) to show the hydrologic timing signal is not event-specific.

This manuscript presents a valuable analysis of compound dry and hot weather (CDHW) events in the Adige River Basin, with the 2022 event serving as a compelling case study. However, to meet the standards of HESS journal, the authors should strengthen the critical synthesis in the introduction, enhance methodological clarity in the abstract, and explicitly discuss the limitations of the data and modeling approach.

---

## Author Response (AR1)

Review of the research article titled "More intense heatwaves under drier conditions: a compound event analysis in the Adige River basin (Eastern Italian Alps)" and submitted to Hydrology and Earth System Science.

**General Comments**

This article presents a study on the impact of climate change on the intensity of compound drought-heatwave (CDHW) events affecting the Adige River Basin, a major basin in the Italian Alps. The authors use E-OBS and ERA5 data to characterize past meteorological conditions, plus two datasets from the Alpine Drought Observatory and the High-Resolution Pan-European Reanalysis to analyse the streamflow of the Adige. Then, they consider an ensemble of 25 EURO-CORDEX RCMs to assess the performance of climate models in reproducing historical trends in CDWH.

The CDWH are defined using daily maximum temperature (TX) and total precipitation (TP) transformed to obtain the SPI-6 index. After detecting all CDWH events since 1950 in ERA5, May 2022 is chosen as a case study, being the second most intense event after 2003, and the most intense in the last 15 years. To study trends in temperature and precipitation associated with this type of event, flow analogs based on 500 hPa geopotential height anomalies (Z500) are adopted. This allows to condition the analysis of the relevant meteorological parameters to the synoptic configuration that produced the specific event, and is a popular strategy to study (compound) meteorological and climatological extremes. For comparison, unconditional trends are also considered.

As it is often done in this type of study, the 1950-2022 reanalysis period was divided into a counterfactual (1951-1980) and a factual (1992-2021) periods, and changes in Z500, SPI-6, TX, TP are quantified between the two periods. Changes in quality of the analogs, annual analog frequency, seasonal distribution are also assessed, together with a simple evaluation of the influence from two large-scale natural variability modes. The analysis is then repeated using EURO-CORDEX simulations, extending the historical period to 2021 using RCP8.5 simulations.

The research question is interesting and coherent with current concerns about the impact of anthropogenic climate change on extreme events, and is well within the scope of HESS. The paper reads very well, all parts are well structured and clearly presented, and the title clearly states the scope of the study. Conclusions are clear and non trivial, and sustained by well described results, obtained with overall robust methodology. The results about the EURO-CORDEX performance (or lack thereof) are particularly interesting and useful, since great attention is being given to the future evolution of extreme events under global warming scenarios, and understanding how reliable models are for specific types of event is important. Also, figures are overall easy to read and well support the interpretation of results.

I found the paper to be overall almost ready for publication, however I would like to point to some minor issues that the authors should be able to address quite easily, detailed in the next section.

Very minor linguistic imprecisions are present here and there, I will only point out one or two that might affect text clarity, the rest should be addressed by the editors.

We would like to sincerely thank the reviewer for the time and effort dedicated to reading and evaluating our manuscript, as well as for the constructive and insightful comments provided. We highly appreciate your positive feedback on the overall quality, clarity, and relevance of our work. We have carefully addressed each of the points raised in your review, and we believe that your suggestions have helped us to further improve the clarity and robustness of the manuscript.

Below, we provide detailed responses to each comment, indicating the changes made accordingly in the revised version of the manuscript.

**Specific Comments**

**Statistical testing**

This comment is general and concerns the execution and presentation of statistical tests used to assess differences in the distribution of several variables.

Overall, I advise against reporting significance at different levels using starts. Although this is a common practice, classic statistical inference is structured around the idea of controlling the probability of type 1 error - the level of the test - and the latter should be specified before running the experiment. I suggest either fixing the level at alpha = 0.05 and reporting only significance at this level, or dropping the significance reporting and directly showing the p-value as in Figure 6. This particularly concerns Fig. 5 and 7a.

Thank you for your suggestion. We have just reported the significance at 0.05 level by using an asterisk symbol (\*), and "ns" when there is no statistical significance. Please see the new version of these figures.

All differences throughout the paper are tested using a two-sample Cramer von Mises test. This test compares the integral difference between the empirical probability distribution functions of the two samples, and it is therefore adequate to assess overall changes in the distribution. However, most of these tests aim at testing a shift in the distribution. While in cases such as in Figure 5 the resulting significant differences are clearly due to a shift, in other occasions this is not the case. In particular, the analog quality in Fig. 6a results significantly different between the two periods, however this difference is very likely due to a change in variability, not a shift; the authors comment on this result stating that the circulation becomes more common, but I would say that this is not the case. To solve this, I suggest to use a different non-parametric test focused on the central tendency of the distribution, and not on the comparison of the entire distribution functions. A good alternative could be the Mann-Whitney U test.

Thank you very much for this thoughtful comment and for suggesting the Mann-Whitney U test as an alternative. Following your advice, we have repeated the comparison of the analog quality using the Mann-Whitney U test to focus on differences in the central tendency of the distributions (Please, see figures 5, 6 and 7) The results confirm that there are no substantial changes in the central tendency between the two periods; the few significant differences detected across the paper are indeed mostly related to changes in variability, as you correctly pointed out. The most relevant differences actually arise from applying the detrending procedure consistently throughout the analysis, rather than from the choice of test itself.

**Section 3.3.1**

While there are several ways to use analogs to perform detection or attribution, the authors here follow the method proposed by Jezequel et al. (2018). In this case, analogs are used to perform a simple stochastic weather generation of events analogs to the May 2022 CDHW, which is not the only possible way to use analogs in general. The methodology is very briefly summarized at lines 186-190. I would suggest adding a few words to explain the technique, possibly using a bullet-point list of the steps, since this explanation could be a little hard to grasp for someone who is not used to the technique.

We thank the reviewer for this helpful suggestion. We have revised the methodology section to include a concise bullet-point list that clearly summarizes the main steps of the analogue-based attribution framework, following Jézéquel et al. (2018) and adapted to our case study. This addition provides a clearer overview of the procedure for readers less familiar with the analogue technique, while full details are kept in the main text. Please, see lines 249-260.

At the end of the section, the evaluation of natural variability is explained, and some limitations are stated. This method -comparing the distributions of the indices in correspondence of the analogs in the two periods - is currently being used not only in other studies, but also in rapid attribution projects (e.g. ClimaMeter). I would say that the main limitation is that, even if significant changes are (or not) observed, it is very difficult to make any statement about the causal relationship or even the correlation between this difference and those observed in the meteorological variables, especially for the AMO, that has a very

long typical period. I suggest explicitly cautioning the reader about the fact that looking at teleconnection indices conditional to the circulation is not the same as looking at changes in circulation or impact variables caused by these teleconnections.

Thank you very much for this helpful comment. We have now added a note of caution in the manuscript to make it clear that comparing the distributions of teleconnection indices conditional on analogue circulation should not be interpreted as evidence of a causal relationship with changes in circulation or impact variables. We also stress that this caveat is especially important for the AMO, given its long periodicity. These clarifications have been included at the end of the section on natural variability (See L.270-274).

**Section 4.2**

Line 255, Figure 4: a clear trend in Z500 is found. However, as also stated later in the article, thermal expansion due to global warming directly causes an increase in geopotential heights. This does not equate observing a change in the circulation, since this is a global trend; Z500 should be detrended before performing this analysis, with previous studies suggesting to use a cubic rather than linear trend specification.

Thank you for pointing this out. Following your suggestion, we have now detrended the Z500 fields before performing the analogue analysis. This is now highlighted in section 3.1 (L.224-225). All reconstructions were repeated with detrended Z500, ensuring that the analogue selection is not biased by the global thermal expansion signal. The results changed slightly after detrending, but the overall outcomes and conclusions remain similar and unaffected. In any case, we kindly suggest you review the results in sections 4.2 and 4.3.

**Minor comments**

Line 20: the factual period is stated to be 1991-2020, while in the rest of the paper is

1992-2021.

**Done.**

Lines 112-113: "...intense precipitations in autumn are mainly driven by cyclonic storms, and in spring/summer are due to snow melt processes, leading to a pluvial regime with two streamflow peaks..." I might be missing something as I am not particularly expert in hydrology, but this sentence seems to state that snow melting is the cause of intense precipitation in the warm season. The authors probably mean that the two streamflow peaks are respectively due to strong cyclonic precipitation in autumn and snow melt processes in spring/summer, this sentence should be corrected.

Thank you for spotting this imprecision. We have rephrased the sentence accordingly to avoid misunderstanding (lines 131–132).

Line 247: "...we randomly reconstructed the atmospheric configurations [...] based on the closest 20 analogs..." I would rather say "we reconstruct the atmospheric configurations [...] using a stochastic simulation based on random sampling from the closest analogs". While there is a random component, most of the work is done by the analogs selection which is deterministic.

Thank you for your recommendation. The text has been revised accordingly (see L.303-305 for details).

I have carefully examined the manuscript titled "More intense heatwaves under drier conditions: a compound event analysis in the Adige River basin (Eastern Italian Alps)" (egusphere-2025-1347-2), submitted by Marc Lemus-Canovas, Alice Crespi, Elena Maines, Stefano Terzi, and Massimiliano Pittore from the Center for Climate Change and Transformation at Eurac Research, Bolzano-Bozen, Italy. This study investigates the increasing intensity and impacts of compound drought and heatwave (CDHW) events in the Adige River basin, with a particular focus on the significant event of May 2022. The authors employ a ranking of CDHW events from 1950 to 2023 using E-OBS data, a flow-analogue attribution approach with ERA5 geopotential height data, and an evaluation of EURO-CORDEX simulations to assess historical changes and future projections. Below, I provide my critical comments and recommendations to enhance the scientific rigor, clarity, and contribution of this work for publication in HESS journal.

We sincerely appreciate the time and effort you have dedicated to reviewing our manuscript. Your constructive comments and suggestions have helped us improve the clarity and robustness of our study. Below, we provide a detailed response, where our revisions and clarifications are highlighted in bold.

1) The abstract and introduction effectively outline the problem, highlight the 2022 CDHW event, and introduce the attribution methodology, making it accessible to a broad audience. The transition from the abstract to the introduction lacks fluidity. The abstract references a ranking of 119 events and the selection of the 2022 event but omits details on the ranking methodology or the rationale for the 1950-2023 timeframe, leaving a disjointed narrative.

Thank you for your comment. We have addressed your suggestion by clarifying in the abstract that the ranking was based on a composite index derived from SPI-6 and daily maximum temperature (TX), thus improving the continuity between the abstract and the introduction (See L 14-16). All the other details on the composite index are provided in the methodology section (see section 3.2). We have also modified the goals of the paper listed at the end of the introduction by clarifying that i) we used the composite indicator and available data to identify the CDHW events occurred over the past decades and ii) based on this list of events we first assessed the relative intensity of the 2022 hot and dry episode in order to motivate the choice of selecting it as a meaningful event for performing the attribution analysis. Regarding the timeframe (1950–2023), this choice is determined by the availability of the E-OBS dataset, which provides consistent daily temperature and precipitation observations since 1950. All the related details to the data used are explained in section 3.1.

2) The abstract briefly mentions the use of E-OBS data, a composite index, and the flow-analogue attribution approach with ERA5 data, but it lacks specificity. For instance, what components constitute the composite index (e.g., temperature, precipitation, spatial extent)? How was the 1-4°C increase in heatwave intensity determined? Providing a brief methodological outline would enhance transparency and allow readers to assess the robustness of the findings upfront.

We appreciate your suggestion to provide greater specificity on the composite index and the attribution methodology. In the revised abstract, we now clarify that the index combines SPI-6 and a heatwave definition based on daily TX (as stated in the reply to your first comment). Regarding the reported 1-4 °C increase in heatwave intensity, this is already clarified in the current version of the abstract as stemming from the analogue-based attribution analysis (please see L23-25), which is described in detail in the Methods section (3.3). Moreover, as we specified above that the heatwave definition uses TX, it should be clearer now that the increase in heatwave intensity refers to the increase in the corresponding maximum temperature. We believe these details ensure the necessary balance between conciseness in the abstract and full transparency in the manuscript.

3) The discussion of atmospheric circulation patterns (e.g., subtropical ridge, warm air from northern Africa) is informative but lacks quantification. Terms like "prolonged periods" and "pronounced precipitation deficits" are vague without supporting data or references to specific magnitudes observed in the Adige basin.

Thank you again for your comment. We have expanded the introduction to clarify the mechanisms sustaining extreme temperatures, highlighting the role of persistent subtropical ridges over southern Europe, warm air advection from northern Africa, and subsidence-driven adiabatic warming. See our integrations in L47-48. This provides a clearer synoptic perspective of the phenomenon before narrowing the focus to the Adige basin. As regards the "qualitative" terminology used, we prefer to keep it general as we refer here to conditions potentially affecting different areas in the central Mediterranean region, so the specific magnitudes of intense heatwaves and precipitation deficits can vary depending on the local climate. Details on specific episodes and regional conditions can be found in the examples reported in lines 54-56.

4) The introduction highlights the scarcity of attribution studies at the catchment scale and the unexplored performance of EURO-CORDEX models, which is a strong motivation. However, it does not preview the specific attribution method (flow-analogue approach) or the limitations of EURO-CORDEX simulations (e.g., spatial resolution, parameterization), which are critical for setting expectations.

Thank you for this valuable suggestion. We have now revised the introduction to explicitly preview both the attribution method applied in this study and the limitations of EURO-CORDEX simulations. Specifically, we included a paragraph that highlights the emergence of flow-analogue approaches as a powerful tool to attribute extreme events, citing their successful application to different types of extremes (lines 66-74). In addition, we expanded the discussion on EURO-CORDEX by acknowledging their added value in complex topography compared to CMIP5/6 simulations, while also stressing their inherent limitations, such as the spatial resolution (0.11°-0.44°) and the uncertainties related to convection and land-atmosphere parameterizations, which are particularly relevant in Alpine catchments (lines 83-87).

5) The use of E-OBS and ERA5 data is mentioned, but their resolution and potential biases (e.g., E-OBS's coarse grid in mountainous areas) are not addressed. This is particularly relevant given the Adige basin's complex topography.

Thank you for pointing this out. We have now explicitly acknowledged in the manuscript that the 0.1° resolution of E-OBS and ERA5-Land may introduce biases when representing local-scale variability, particularly given the complex topography of the Adige basin. To address this, we have incorporated a reference that discusses these limitations, providing further support to our statement (Bandhauer et al., 2021). Nevertheless, these datasets remain the only long and continuous daily gridded records of both temperature and precipitation available for the region, which makes them the most suitable choice for our analysis. Please, see L. 150-155.

6) The introduction cites several studies (e.g., Viviroli et al., 2007; Hao et al., 2022) to establish the importance of the Alpine region and compound extremes but lacks a critical synthesis. For example, it does not address whether previous studies have underestimated snow dynamics or elevation-dependent warming in the Alps, which are highlighted as unique challenges. I strongly

recommend considering these two studies: Assimilation of sentinel-based leaf area index for modeling surface-ground water interactions in irrigation districts; Elevation dependent change in ERA5 precipitation and its extremes.

Thank you very much for this valuable suggestion. We fully agree that snow dynamics and elevation-dependent warming pose unique challenges in alpine attribution studies. Indeed, in the revised Introduction we explicitly highlight these processes as key drivers shaping CDHW events in mountain catchments (lines 73-80), together with the interrelated role of atmospheric water demand and water availability. These aspects are already discussed in the context of the limited representation of alpine processes in large-scale studies, supported by references such as Brunner et al. (2023), Jenicek et al. (2016), Pepin et al. (2015), Van Loon et al. (2015), and Mastrotheodoros et al. We believe this already provides the necessary critical synthesis, while keeping the introduction concise and directly focused on the objectives of the study. Regarding the two additional references suggested: the first one, focusing on leaf area index assimilation for irrigation modeling, is outside the scope of our study. The second one investigates elevationdependent changes in precipitation and extremes across several regions globally. While highly relevant in a broader context, our focus here is on hot and dry events in a relatively small portion of the Alpine region. For this reason, we prefer to cite elevationdependent warming more generally as a key local factor to be considered when extrapolating the findings of large-scale studies to the Alpine context.

7) The abstract's note that over half of the EURO-CORDEX models failed to reproduce observed changes suggests potential issues with model selection or validation. The introduction does not foreshadow this, which could undermine confidence in the projections.

We understand the reviewer's concern. However, the finding that more than half of the EURO-CORDEX models fail to reproduce the observed changes is in fact one of the main results of our study, rather than a limitation of model selection or validation. For this reason, we believe it is more appropriate to present and discuss this issue in detail in the Discussion sections (see lines 536-562), where we highlight its implications for confidence in future projections. Introducing this aspect in the Introduction would risk anticipating key results and affect the narrative flow.

8) The streamflow story leans on one gauge (Trento) plus HERA; June reductions are attributed largely to earlier snowmelt. Please (i) discuss/quantify confounding from irrigation/hydropower operations (not just note restrictions), (ii) report whether HERA is "naturalized" or includes management, and (iii) add a simple basin water-balance perspective (snow cover, PET/ET0, soil moisture) to separate supply vs. demand effects.

Thank you for your suggestions. On point (i), We acknowledge the reviewer's point on the potential confounding effects of irrigation and hydropower management. Most of the largest dams and artificial lakes in the Adige basin (e.g., Santa Giustina, 182 Mm³, built in 1951; Resia, 120 Mm³, built in 1949; Stramentizzo, 11.5 Mm³, built in 1956) were already in place at the beginning of the study period. This suggests that the major hydropower infrastructure was largely stable during 1951–2020, even though operational strategies may have evolved over time. We have included these details in the data section (see L172-178). In addition, quantifying these changes in management practices is not straightforward and would require additional datasets that are beyond the scope of this study.

As for irrigation, we agree that it represents another potential confounding factor. However, assessing its long-term influence would require consistent proxies (e.g., ET or soil moisture anomalies) or dedicated modelling exercises to reconstruct agricultural water use from 1951 onwards which goes beyond the scope of the study.

For the second point (ii), we have clarified in the revised manuscript that HERA is not a naturalized dataset, as its long-term discharge trends reflect not only climate variability but also the influence of human management. However, as we pointed out before, the major hydropower infrastructure was largely stable during 1951–2020. In any case, we have highlighted that direct links between climatic drivers and discharge trends should be interpreted with caution (See L. 177-178).

Regarding the third point (iii), we acknowledge the importance of a basin-scale water-balance perspective to disentangle supply- versus demand-driven effects. Indeed, variables such as snow cover, PET/ET0, and soil moisture would provide valuable complementary insights. However, quantifying these processes in a consistent way for the entire catchment would require a dedicated hydrological modelling framework, which goes beyond the scope of this study. Instead, our analysis focuses on the meteorological drivers and their link to streamflow changes, while recognising that water balance components and management practices also play a role in shaping the observed discharge response (lines 172-178).

9) You show earlier snowmelt and an April/May discharge bump followed by June deficits. Consider cross-checking with independent snow data (in situ SWE, satellite snow extent) and add confidence intervals for the reported "30–40 cm per 30y" and "±40–60 m³/s" trends.

Thank you for this helpful suggestion. We have added a new supplementary figure showing the decrease in snow-covered days from MODIS observations, separated by elevation ranges, to support the results on earlier snowmelt (see Fig. A3). These results are clearly consistent with what we have already shown through the 3 in-situ observations used in this work. Refer to L415-417 for supporting text.

Figure A3. a) Average snow cover percentage for the period 2000–2023 by elevation band in the Adige catchment, derived from MODIS data. (b) Decadal trends in snow cover percentage over the same elevation bands. Cells highlighted with a black border denote trends that are statistically significant at the 95% confidence level.

In addition, we now provide confidence intervals at the 95% level for the long-term trends in both snow depth and river discharge, which have been incorporated into the main text (see lines 415-421). To further illustrate these uncertainties, we also include two supplementary figures displaying the confidence bands for both variables (Fig. A2).

Figure A2. (a) Snow depth at three historical observatories in the northwestern Adige catchment (Diga di Gioveretto, Fontana Bianca, and Roia di Fuori) with 30-year day-of-year trends and their 95% confidence intervals over 1981–2018. (b) Same as (a) but for Adige River discharge at the Trento gauge station.

10) Beyond sign/magnitude counts, include formal skill scores (bias, RMSE, correlation, CRPS) for both conditioned and unconditioned reconstructions, and try simple emergent-constraint or performance-based weighting to see if an informed subset reduces the underestimation. Clarify implications of stitching historical with RCP8.5 to 2021.

Thank you for these valuable suggestions. The primary objective of our study regarding the EURO-CORDEX analysis was to assess the sign and magnitude of change (as stated in the objectives), rather than to provide a full forecast-verification suite. That said, we have taken up your idea of using performance-based selection. Specifically, we identify a subset of "better-performing" EURO-CORDEX models based on the similarity (lower RMSD) of their Z500 analogue patterns to observations (ERA5), consistent with the circulation-conditioned framework (see Fig. A3 for Z500). We now also provide the same evaluation for TX (new Fig. A4). Using the top-5 models as an informed subset does not significantly reduce the underestimation in the reconstructions of Z500 and TX -the two variables directly conditioned on circulation- so our conclusions are robust to model selection. We therefore keep the results centered on the sign-magnitude framing and the analogue-based evaluation, while documenting the performance-based subset test as noted above.

On the stitching of historical and RCP8.5 runs through 2021, we clarify that -consistent with other studies employing EURO-CORDEX- using only the early years of an RCP scenario (typically RCP8.5) does not introduce major artefacts in the results. Moreover, RCP8.5 best reflects the current pace of warming. This justification has been added in the Methods (lines 162-164).

11) Where you claim significant changes, consistently show effect sizes with Cls. You use Cramér–von Mises for some tests; extend uncertainty quantification to the event ranking, severity composites, and the discharge change maps (e.g., bootstrap over analogues and spatial blocks).

We appreciate this valuable comment. Following your suggestion, we have extended the uncertainty quantification across all analogue-based reconstructions. Specifically, the discharge change maps now also reflect the statistical significance of the observed differences (see new Figure 8).

Figure 8. Relative changes in river discharge (%) in the Adige River basin between the periods 1992–2021 and 1951–1980 for April, May, and June. Black dots indicate statistically significant differences at the 95% confidence level. ERA5-Land pixels with

Regarding your earlier point, we have also broadened the confidence intervals as described in our response to Comment 9.

As for the event ranking and severity composites, these are derived directly from metric calculations (see methods section). Since they are not based on sampling or resampling procedures, no meaningful uncertainty analysis can be applied in this case.

12) Since analogues are conditioned on one pattern, stress-test conclusions by repeating the pipeline for another major CDHW (e.g., 2003/2018) to show the hydrologic timing signal is not event-specific.

Thank you again for your comment. Indeed, the May-June 2003 event was also one of the most intense CDHWs over the Adige catchment (ranked 2nd). As often happens, the strongest events are driven by similar circulation conditions. Therefore, we checked whether our analyses included analogues from May-June 2003 (Fig. R1). As you can see, some analogues are indeed taken from those months in 2003. We have clarified this in L. 514–517, noting that our results can also be extrapolated to events such as 2003, given the strong similarity in the underlying mechanisms.

Fig R1. Number of analogues of the May 2022 event detected by year.

Although the 2018 event was very impactful in central and northern Europe (especially in France, Belgium, and the Netherlands), it was not as intense in the Adige catchment and does not reproduce exactly the same circulation pattern associated with extreme heat events in our target region. That said, replicating the same approach for 2003 would likely lead to results similar to those obtained for 2022, since the circulation does not change substantially from one event to the other (Fig R2).

Fig. R2. Averaged Z500 over the May 2003 and 2022 episodes.

This manuscript presents a valuable analysis of compound dry and hot weather (CDHW) events in the Adige River Basin, with the 2022 event serving as a compelling case study. However, to meet the standards of HESS journal, the authors should strengthen the critical synthesis in the introduction, enhance methodological clarity in the abstract, and explicitly discuss the limitations of the data and modeling approach.

We sincerely thank the reviewer for this constructive and encouraging overall assessment. We have aimed to address all comments and suggestions in detail, and we believe these changes have significantly improved the manuscript.

---

## Author Response (AR2)

I have read your manuscript again and confirmed that you have addressed all reviewers's comments.

According to the provided reviews with in-depth analysis about your manuscript, and your timely and detailed responses, I do not think we need to go back into a second review round.

I have a minor correction proposal:

Please review Figure A1: It does not have the letters ((a)- (j)) indicating which ones are the corresponding panels within the figure.

Thank you very much for your positive feedback. We have implemented the suggested correction, adding the panel labels ((a)–(j)) to Figure A1 in the newly uploaded version of the manuscript.

Best regards,

Marc Lemus-Canovas on behalf of the authors